# Anti-Stokes photoluminescence probing k-conservation and thermalization of minority carriers in degenerately doped semiconductors

K. Mergenthaler[1], N. Anttu[1], N. Vainorius[1], M. Aghaeipour[1], S. Lehmann[1], M.T. Borgström [1], L. Samuelson[1] & M.-E. Pistol[1]

It has recently been found that anti-Stokes photoluminescence can be observed in degenerately n-doped indium phosphide nanowires, when exciting directly into the electron gas. This anti-Stokes mechanism has not been observed before and allows the study of carrier relaxation and recombination using standard photoluminescence techniques. It is important to know if this anti-Stokes photoluminescence also occurs in bulk semiconductors as well as its relation to carrier recombination and relaxation. Here we show that similar anti-Stokes photoluminescence can indeed be observed in degenerately doped bulk indium phosphide and gallium arsenide and is caused by minority carriers scattering to high momenta by phonons. We find in addition that the radiative electron-hole recombination is highly momentum-conserving and that photogenerated minority carriers recombine before relaxing to the band edge at low temperatures. These observations challenge the use of models assuming thermalization of minority carriers in the analysis of highly doped devices.

---

[1] Department of Solid State Physics and NanoLund, Lund University, Box 118, 221 00 Lund, Sweden. Correspondence and requests for materials should be addressed to K.M. (email: kilian.mergenthaler@ftf.lth.se) or to M.-E.P. (email: mats-erik.pistol@ftf.lth.se)

Anti-Stokes photoluminescence, where photoluminescence occurs at energies higher than that of the excitation photons, can occur through different physical mechanisms[1] and is of significant technological importance for diverse research fields ranging from medicine and biology to optoelectronics and photonics[2–6]. For biological imaging, anti-Stokes photoluminescence allows for improved sensitivity with reduced photo damage[2,7]. Lasers using this effect have been demonstrated[4,8]. Anti-Stokes photoluminescence enables enhanced detection of infrared photons[3], which can be used for white light generation[9] and could improve the efficiency of solar cells[5].

Anti-Stokes luminescence has, in semiconducting materials, been observed in zero-dimensional systems such as quantum dots;[10,11] in two-dimensional systems such as type II heterojunctions[12], CdS nanoribbons[13], and GaAs quantum wells;[14] and in high band gap bulk materials such as GaN[15] and ZnO[6,16].

Most reported anti-Stokes photoluminescence in semiconductors requires states below the band edge and excitation by Auger processes[12,17,18] or two-photon absorption (photon upconversion)[6,11,15,16,19,20]. Two other anti-Stokes mechanisms involve phonons. In one, the simultaneous absorption of photons and phonons enables absorption below the band gap[21,22] and can lead to upconverted band edge luminescence. Alternatively, photogenerated charge carriers can be excited to higher states through phonon absorption, which can lead to recombination involving these higher energetic excited states[23].

For efficient anti-Stokes photoluminescence, two-step processes with an intermediate state are advantageous, due to the (typically) lower probability of processes involving interaction between a larger number of particles. In bulk semiconductors, two-step anti-Stokes photoluminescence requires defect states within the band gap. We recently reported on anti-Stokes photoluminescence in degenerately n-doped InP nanowires[24,25], where the anti-Stokes mechanism relies on absorption of photons with energies less than the Fermi energy of the degenerately doped material, followed by phonon scattering of the photoexcited hole to higher $\mathbf{k}$-values (where $\mathbf{k}$ is the charge carrier momentum). Such anti-Stokes photoluminescence has so far not been reported for bulk semiconductors.

In this work we demonstrate that such phonon assisted anti-Stokes photoluminescence is not limited to nanoscale materials, but is a more general property of direct band gap semiconductors. During this experimental demonstration of anti-Stokes photoluminescence, we also identified important aspects of $\mathbf{k}$-conservation in radiative recombination as well as thermalization of photogenerated minority carriers. We find that the radiative recombination in our samples is strongly dominated by $\mathbf{k}$-conserving vertical transitions, despite high doping. In contrast, in the literature it is often assumed that the $\mathbf{k}$-selection rule is violated in strongly doped semiconductors. We will argue that the photoluminescence lineshape is determined by the distribution of holes recombining vertically (conserving $\mathbf{k}$) with the Fermi sea of electrons. In addition, we find that the charge carriers recombine before the holes have thermalized to the valence band maximum. This means that the photoluminescence lineshape is not determined by holes that have relaxed to the valence band maximum before recombining, which would lead to photoluminescence peaked at the band gap.

## Results

**Anti-Stokes photoluminescence in bulk semiconductors.** Anti-Stokes photoluminescence in highly doped semiconductors was first observed in sulfur-doped InP nanowires[24,25]. It was interpreted as a nanowire-specific attribute due to the enhanced coupling of charge carriers to optical phonons in nanoscale

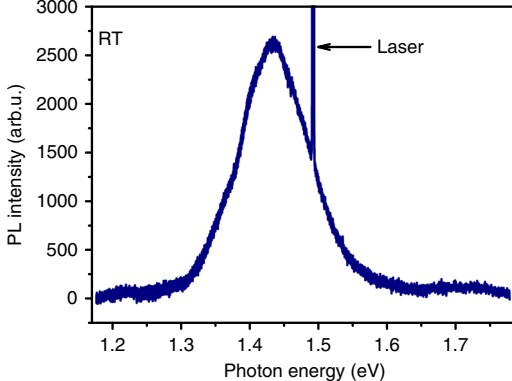

**Fig. 1** Room temperature photoluminescence of indium phosphide. The indium phosphide is sulfur doped, and for excitation at 1.494 eV (830 nm), we observe photoluminescence on both the low-energy and high energy side of the exciting laser line, corresponding to Stokes and anti-Stokes photoluminescence, respectively

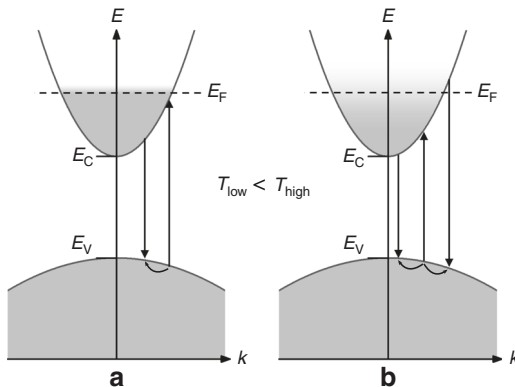

**Fig. 2** Absorption and radiative recombination in a semiconductor. The semiconductor is degenerately n-doped, has a direct band gap, and the transitions are $\mathbf{k}$-conserving. Here, **a** is at low temperature and **b** is at high temperature

materials[13] and efficient heating of the electron gas by the exciting laser. However, the results presented in Fig. 1 show that anti-Stokes photoluminescence can also be observed in degenerately n-doped bulk InP. The room temperature band gap of intrinsic InP is 1.35 eV[26] and the Fermi energy of the sample was estimated to be 1.50 eV. For excitation with 1.494 eV photons at room temperature, we observed luminescence on both sides of the laser energy. Similar results as for n-doped InP were also found for n-doped and p-doped GaAs indicating that our results are quite general. Despite an extensive literature search, we have not found any previous reports on this anti-Stokes effect, which is slightly surprising since the experiment can be performed on commercially available samples.

A schematic image of our model of the anti-Stokes photoluminescence mechanism in degenerately n-doped direct band semiconductors is shown in Fig. 2. The gray shaded areas depict states filled with electrons and the vertical arrows depict absorption and emission of a photon. Figure 2a shows the situation for low electron gas temperatures, where photon absorption is forbidden for photon energies far below the Fermi energy ($E_F$) since there are no available empty electron states. For example at 4 K, 5 meV below the Fermi energy less than a fraction of $10^{-6}$ of the states are unoccupied, and 98% of the states are occupied at $4k_BT = 1.4$ meV below the Fermi energy. Figure 2b shows the situation at higher electron temperature, where the

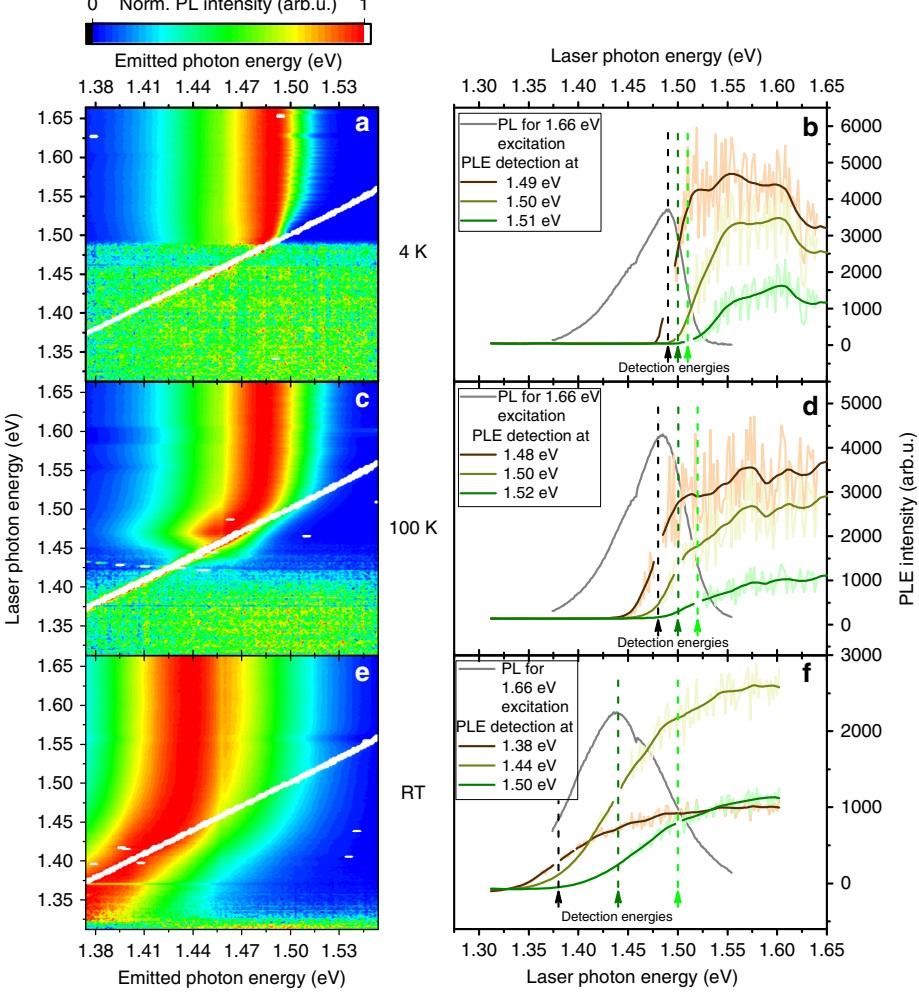

**Fig. 3** Excitation energy dependence of the photoluminescence. The samples are n-doped indium phosphide. The figure shows photon energy contour plots measured at 4 K (**a**), at 100 K (**c**), and at room temperature (**e**). The excitation laser energy is visible as a diagonal white line. **b**, **d**, **f** show photoluminescence excitation spectra for selected detection energies at 4 K, 100 K, and room temperature, respectively, together with the photoluminescence spectrum for 1.66 eV at the respective temperature (in gray). The light colored spectra show the raw data and the dark colored spectra the smoothed data. The detection energy of the photoluminescence excitation spectra is indicated by vertical arrows

elevated electron gas temperature allows absorption at much lower energies (at 300 K, $4k_BT = 103$ meV). A high electron gas temperature can be the result of a high sample temperature or due to direct heating of the electron gas by the photo-exciting laser[25].

We show, later in this work, strong evidence that the radiative recombination in our samples is strongly **k**-conserving, and due to this **k**-conservation, anti-Stokes photoluminescence requires scattering of the photo-excited hole to higher **k**-values. The strong temperature dependence of most hole scattering processes[27] leads to negligible **k**-increasing scattering rates at low temperatures. This means that at low temperature, Fig. 2a, there is negligible scattering of the photo-excited hole to higher **k**-values and thus, no observed anti-Stokes photoluminescence. In Fig. 2b the elevated sample temperature allows hole scattering to higher **k**-values and anti-Stokes photoluminescence can be detected. Hole scattering to lower **k**-values leads to Stokes photoluminescence, which may be detected at both temperatures shown in Fig. 2.

**Anti-Stokes photoluminescence in degenerately doped n-InP.** In order to get further information on the processes behind the anti-Stokes photoluminescence, we measured the excitation energy dependence and the temperature dependence of the anti-

Stokes photoluminescence in degenerately n-doped InP. The results at 4 K, 100 K, and room temperature are presented as photon energy contour plots in Fig. 3a, c, e, respectively. Note that a horizontal line cut corresponds to one PL measurement (and the shown spectra are normalized to the luminescence maximum for such cutlines). A vertical cutline in the raw data in turn corresponds to a photoluminescence excitation spectroscopy (PLE) measurement, if considered before normalization. Due to normalization to the luminescence maximum, the excitation laser energy is visible in Fig. 3 as white diagonal lines and luminescence below the diagonal white line is equivalent with anti-Stokes photoluminescence.

For the measurements at 4 K (Fig. 3a) luminescence is only detected for laser photon energies higher than about 1.48 eV (while the intrinsic band gap energy, $E_{g,intrinsic}$, is 1.424 eV at 4 K). For laser photon energies lower than 1.48 eV, we do not observe any Stokes or anti-Stokes PL. We conclude from this lack of PL signal that for laser energy < 1.48 eV, the band-to-band absorption is negligible. Note that for n-doped InP nanowires, the situation was different and absorption was possible into the electron gas also at a sample temperature of 4 K[24] giving rise to Stokes PL. The low-energy absorption in nanowires was attributed to laser-induced heating of the electron gas.

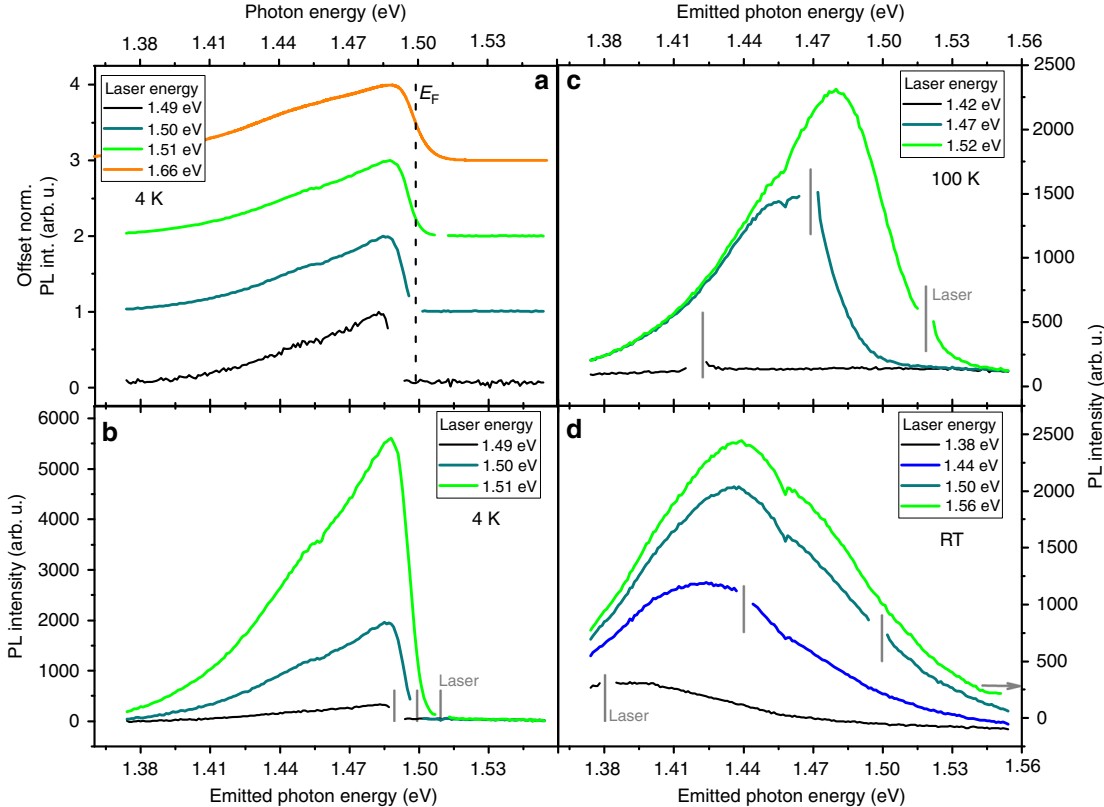

**Fig. 4** Photoluminescence spectra of n-doped indium phosphide. The spectra in **a** are measured at 4 K using different excitation energies. These spectra are normalized. The vertical black line indicates the doping induced Fermi energy estimated from the photoluminescence for 1.66 eV excitation. In **b–d**, the spectra are measured at 4, 100, and 300 K, respectively. These spectra are not normalized. The respective excitation laser energy is indicated by vertical gray lines

For measurements at 100 K, both anti-Stokes and Stokes PL is visible for laser energies between about 1.44 and 1.48 eV (Fig. 3c), indicating that absorption below the Fermi energy is possible as expected. At room temperature, luminescence can be seen for excitation energies down to about 1.32 eV (Fig. 3e).

**Photoluminescence excitation spectroscopy.** An established method to study absorption in weakly absorbing semiconductors is PLE, where the dependence of the PL intensity on excitation energy is measured. Figure 3b, d, f show such PLE spectra for selected detection energies at 4 K, 100 K, and room temperature, respectively. Our PLE spectra cover both the anti-Stokes regime and the Stokes regime, in contrast to conventional PLE measurements, which almost exclusively only cover the Stokes regime.

The shift of the on-set of the PLE spectra with temperature confirms that high temperatures allow excitation deep below the Fermi energy. It is important to remember that, at the selected detection energies, PLE measures only inter-band absorption that leads to the creation of holes in the valence band. Inter-subband absorption, free-carrier absorption, and other similar absorption processes are not detected. The gray spectra in Fig. 3b, d, f are normalized PL spectra excited at 1.66 eV.

In Fig. 3c, e we show that for absorption below the Fermi energy, the emission peak energy shifts to lower energies with decreasing excitation energy until the sample stops absorbing. The effect is also visible in the PL spectra shown in Fig. 4. This shift is caused by a reduced emission on the high energy side of the PL spectrum when the laser energy decreases. The gradual decrease of absorption for decreasing excitation energies reduces the overall PL intensity, but the reduction is stronger for anti-Stokes PL than for Stokes PL. Note that the situation was different

in nanowires, where no spectral change of the anti-Stokes peak was observed as a function of laser energy[24].

**Evidence of k-conservation in radiative recombination.** It has been discussed in some cases that in degenerately n-doped semiconductors, photo-excited holes relax/thermalize to the top of the valence band and then recombine with an electron from the electron gas with a possible change in **k**-vector[28]. The relaxation of the **k**-selection rule is then caused either by crystal imperfections introduced by dopants or by momentum taken up by the electron gas[29]. The photoluminescence lineshape is then determined by the occupancy of the electron states and the transition matrix elements between holes at the top of the valence band and the electrons[29].

In our samples the situation is very different. The **k**-selection rule is strongly obeyed for radiative recombination. We know this since at low temperatures, 4 K, there is no anti-Stokes PL that we can measure when we excite below the Fermi energy, see Fig. 4a. Absorption at 1.49 eV leads exclusively to Stokes PL, although the Fermi energy is at 1.50 eV. (note that the PL peak intensity increases by almost a factor of 3 when increasing the laser energy from 1.50 to 1.51 eV in Fig. 4b. This indicates strongly that the degenerate doping restricts the excitation at 1.50 eV. Therefore, for the holes generated by the 1.49 eV laser, there should be such higher **k** electrons available, and recombination with such electrons should lead to recombination with emission of photons with an energy > 1.49 eV. However, we find no such luminescence in the 1.49 eV excitation spectrum, and hence conclude that **k**-non-conserving recombination, which would lead to anti-Stokes PL, is not significant).

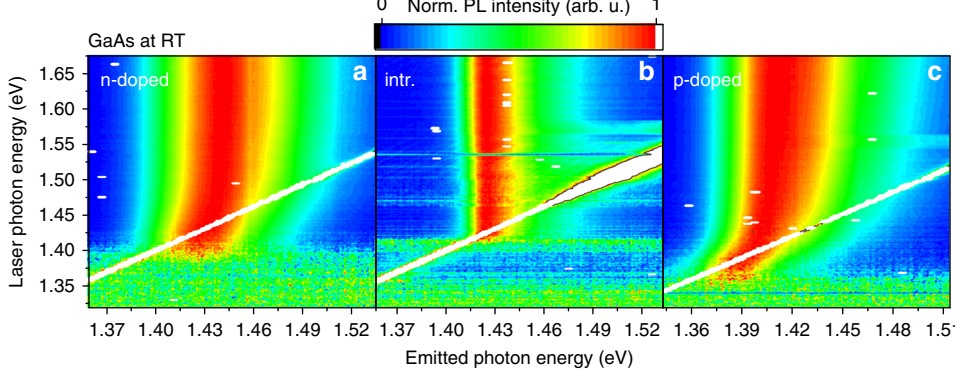

**Fig. 5** Photoluminescence of gallium arsenide. The figure shows photon energy contour plots for **a** n-doped gallium arsenide, **b** nominally undoped gallium arsenide, and **c** p-doped gallium arsenide. The temperature is 300 K and the excitation energy has been varied

Next, we note that there is Stokes PL and, as argued above, the **k**-selection rule is obeyed. We thus know that there is scattering of holes to smaller **k**-values, also at low temperatures.

Importantly, we argue that photo-excited holes recombine with electrons before they have had time to relax/thermalize to the top of the valence band. Note that above, we argued that the **k**-selection rule is obeyed. We also noted that holes can relax toward the top of the valence band. In that case, if the holes scatter all the way to the top of the valence band before recombining, we would expect the photoluminescence to be peaked at the band gap energy. This is not the case in our measurements (see for example Fig. 4c where the luminescence peaks far above the 1.34 eV band gap of bulk InP at RT). Furthermore, we see that the peak position depends on the laser energy as seen for example in Fig. 4c. This dependence of the peak position on the laser energy indicates that the initial state of the hole affects the resulting luminescence. Therefore, complete thermalization of the photogenerated holes has not occurred before luminescence.

Presently, the exact scattering mechanism of the holes is unknown. Either there is only one scattering event before recombination, or there is a set of low-energy scattering events. Furthermore, the scattering can be to bandstates as well as to defect states. This uncertainty prohibits accurate modeling of the emission lineshape. However, even with this uncertainty in the exact scattering and recombination dynamics, the anti-Stokes PL is now possible to depict. At high temperatures, holes scatter from low **k**-values to higher **k**-values causing anti-Stokes PL, as illustrated in Fig. 2b. This process should be temperature activated and this is precisely what we observe in Fig. 4.

Summarizing, our results show that (1) in degenerately n-doped bulk InP, the majority of radiative recombination of photo-excited holes occurs through vertical, **k**-conserving transitions and (2) the recombination occurs before the holes relax/thermalize to the valence band maximum.

**Anti-Stokes photoluminescence in GaAs**. Our interpretation of the observed anti-Stokes PL suggests similar processes in other degenerately doped direct band gap semiconductors. Figure 5 shows the excitation energy dependence of the normalized luminescence of intrinsic and degenerately doped GaAs samples measured at room temperature. Both degenerately doped samples (Fig. 5a n-type doped and Fig. 5c p-type doped) exhibit significant anti-Stokes PL, and for both samples the PL peak energy shifts to lower energies with decreasing excitation energy for excitation into the luminescence peak. Figure 5b shows that for nominally undoped GaAs, the anti-Stokes PL is less pronounced and no significant peak shift is observed.

According to the reasoning for n-doped InP, an asymmetric spectral change relative to the laser energy indicates that the radiative recombination is dominated by **k**-conserving recombinations of hot minority charge carriers.

## Discussion

In summary, we have demonstrated that anti-Stokes PL can be achieved in degenerately doped InP and GaAs, and that the underlying process is scattering of photo-excited minority carriers to higher **k**-values. We argue that the radiative electron-hole recombination is **k**-conserving and that the recombination rate is faster than the **k**-changing scattering rate. Thus, we believe that the spectral PL shape is not determined by indirect (non **k**-conserving) radiative recombinations of Γ-point minority carriers (as commonly assumed), but by the minority charge carrier energy relaxation rate relative to the recombination rate. The similarities of the results of n-type InP, n-type GaAs, and p-type GaAs suggest that our model may be general for all direct band gap III–V semiconductors.

Our findings allow a more accurate description of radiative recombination in semiconductors and show that anti-Stokes PL may influence the behavior of opto-electronic semiconductor devices. Future experiments should investigate the time-dependence of the Stokes and anti-Stokes PL to yield additional insight into the underlying minority carrier scattering and recombination dynamics. Comparing detailed modeling with the PLE and absorption experiments should give important information on the hole scattering dynamics.

Finally, we would like to highlight that we saw in our steady-state experiment a strong indication of a lack of thermalization of photogenerated minority carriers. Such an observation challenges the use of simplified models, based on the thermalization assumption, for describing carrier dynamics in the analysis of semiconductor devices with highly doped regions, such as degenerately doped p+n+-junction-based solar cells. Although our results pertain only to the physics of highly doped bulk InP and GaAs, we note that upconversion has been studied for its device relevance. For instance Auger-assisted upconverted photocurrent has been studied for intermediate band solar cells[30] and photo-chemical upconversion for other photovoltaic devices[31].

## Methods

We studied a widely used commercially available single crystalline bulk substrate grown by the vertical gradient freeze method (VGF), which is a modification of the Bridgman technique[32]. The studied InP sample was degenerately sulfur doped, that is, n-type, with a doping concentration of about $10^{19}$ cm$^{-3}$. The studied GaAs samples were silicon-doped n-type (about $4 \times 10^{18}$ cm$^{-3}$), zinc-doped p-type (about $1 \times 10^{19}$ cm$^{-3}$), and nominally undoped GaAs.

For photoluminescence measurements, the samples were mounted onto the cold finger of a continuous flow liquid helium cryostat system. The samples were

optically excited by a continuous wave (cw) tunable optically pumped Ti:Sapphire laser. The laser light was focused on the sample and the optical power density was kept constant around 40 kW cm⁻². The photoluminescence was collected by a ×50 microscope objective lens, dispersed by a single monochromator, and detected by a scientific Andor Neo CMOS-camera. To allow measurements close to the laser line energy, a cross-polarized dark field PL setup was used[33], where the polarization state of the excitation light is orthogonal to the polarization of the detection. In our setup, we reached a suppression of the backscattered laser line by a factor of about $10^5$ allowing us to measure the laser line simultaneously with the photoluminescence.

**Data availability**. The data that support the findings of this study are available from the corresponding author upon request.

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

## Acknowledgements
This work was performed within NanoLund at Lund University and was supported by NanoLund, the Swedish Research Council VR, and the Swedish Foundation for Strategic Research, SSF.

## Author contributions
K.M. performed the measurements, analyzed the data and wrote the first draft of the manuscript. N.A. helped analyze the data and participated in writing the manuscript. N. V. built the experimental setup used in the study. M.A. performed control absorption measurements. S.L. supplied samples and participated in writing the manuscript. M.T.B. supplied samples. L.S. participated in discussions of the results. M.-E.P. initiated the project, helped analyze the data, and participated in writing the manuscript. All authors have read and approved the manuscript.

## Additional information

**Competing interests:** The authors declare no competing financial interests.

