## [Peer Review File · Nature Communications]

Reviewers' Comments:

Reviewer #1:

Remarks to the Author:

The manuscript presents a study on the anti-Stokes luminescence property of degenerately doped bulk semiconductors. The novelty of this work lies in the observation of anti-Stokes emission from the degenerately doped bulk InP and GaAs semiconductors, which locates at the shorter wavelength side of the emission peak under Ti:sapphire laser excitation. However, there is a fundamental mistake in describing this phenomenon, which is only the one-photon-for-one-photon in-band luminescence process and only the emission at shorter wavelength side belongs to anti-Stokes process, rather than the "photon upconversion", and the other part of the emission is Stokes emission because of the lower energy emitting photons (namely the emission peak wavelength is longer than the pump laser). The concept "photon upconversion" from either lanthanides or semiconductors refers to the photon emission resulted from absorptions of multiple lower-energy irradiation photons, e.g., the typical TPA process for semiconductor. Such one-photon-for-one-photon luminescence phenomenon is also a generally observed process for the resonantly pumped emissions from lanthanides, where a small part of the emission profiles at shorter wavelength side. Therefore the basis of the discussion in the present work is not correct (eg, the descriptions in Fig 1 caption are wrong).

Furthermore, these in-band emission depends sensitively on the temperature, and it is much obvious for observation of the anti-Stokes profile part at RT than the low temperature such as at 100 K or 4K, see Figs 3-4. These data indicate the close association of the emission with the thermalization and the need for phonons in assisting the anti-Stokes emission part. In Figs 3,4 we can also see that the main emission photons have lower energy than the pump laser, further suggesting that it is not photon upconversion.

It is also noted that the authors had observed similar phenomenon in semiconductor nanowires (the refs 24,25), and the present work gives a extensive study on the emission property from bulk. Besides, the work regarding to the demonstration of the k-conservation is acceptable.

In summary, this manuscript failed to meet the level of Nature Communications because of the uncorrected description and discussion on the optical phenomenon, and I would not recommend this manuscript to be published in a more professional journal after a thorough revision.

Reviewer #2:

Remarks to the Author:

The paper provides an important perspective on the phenomena of up-conversion in semiconductor alloys. Previous work is adequately reviewed and the authors present some carefully conducted experiments and make a strong case for their interpretation. I have only a couple of comments that the authors could consider to make the paper clearer and more comprehensive:

- The authors use the terms up-conversion and down-conversion. I personally feel that these terms should correspond to either the energy of two photons being combined to produce one higher energy photon (up-conversion), or the reverse, the energy of one high energy photon being used to produce two lower energy photons. I am somewhat comfortable with the authors use of up-conversion, but not down-conversion which they seem to use to describe the conventional Stokes-shift photoluminescence. Down conversion should not conserve particle number.

Sometimes this is called 'down-shifting' by the photovoltaics community, but to my mind it is simply a Stokes shift in PL. I would suggest the authors remove the cases where down-conversion is used and replace it with Stokes PL.

- The authors have made a good attempt at reviewing the literature. Since their paper is quite general in its appeal, I feel they could reference a little more widely. For inorganic UC systems I

am aware of the following recent papers which they may wish to consider:

D. M. Tex, I. Kamiya, and Y. Kanemitsu, "Efficient upconverted photocurrent through an Auger process in disklike InAs quantum structures for intermediate-band solar cells," *Phys Rev B*, vol. 87, no. 24, 2013.

N. P. Hylton, et al., "Photoluminescence upconversion at GaAs/InGaP2 interfaces driven by a sequential two-photon absorption mechanism," *Phys Rev B*, vol. 93, no. 23, p. 235303, Jun. 2016.

- Finally the authors mention the possibility for using up-conversion for solar energy conversion. The laser intensities used in the experiments here are greatly in excess of any photon density that can be achieved with sunlight so this application does not seem to be compelling. To my knowledge, molecular up-conversion systems appear to be most promising for this application, see for example the recent review:

T. F. Schulze and T. W. Schmidt, "Photochemical upconversion: present status and prospects for its application to solar energy conversion," *Energy Environ. Sci.*, vol. 8, no. 1, pp. 103–125, 2015.

Reviewer #3:

Remarks to the Author:

This paper describes an investigation of photon upconversion in bulk semiconductors. The authors discover an unusual k-conserving transition in the bulk, which they believe general to all direct bandgap III-V semiconductors. The authors also claim that their discovery may influence the behavior of optoelectronic semiconductor devices.

The referee feels this study a bit preliminary. The authors are suggested to verify the phenomena in other materials systems and investigate this effect in a real semiconductor device. The study also looks too specialized to the referee. To date, a number of materials systems including nanocrystals and small molecules are available for upconverting photon energy, inspiring quite a few new applications. This study focuses on a specific physical process in a particular material, and the implications of the investigation are not clear. It may not appeal to the broad readership of nature communications or excite the direct interests of the community. I therefore don't recommend acceptance of the work.

Reviewer 1

We are happy to hear that the reviewer finds our demonstration of k-conservation acceptable, which is one of the key points of the work. The reviewer raises the following technical point:

R1 comment 1: *“The manuscript presents a study on the anti-Stokes luminescence property of degenerately doped bulk semiconductors. The novelty of this work lies in the observation of anti-Stokes emission from the degenerately doped bulk InP and GaAs semiconductors, which locates at the shorter wavelength side of the emission peak under Ti:sapphire laser excitation.*

However, there is a fundamental mistake in describing this phenomenon, which is only the one-photon-for-one-photon in-band luminescence process and only the emission at shorter wavelength side belongs to anti-Stokes process, rather than the “photon upconversion”, and the other part of the emission is Stokes emission because of the lower energy emitting photons (namely the emission peak wavelength is longer than the pump laser).

The concept “photon upconversion” from either lanthanides or semiconductors refers to the photon emission resulted from absorptions of multiple lower-energy irradiation photons, e.g., the typical TPA process for semiconductor. Such one-photon-for-one-photon luminescence phenomenon is also a generally observed process for the resonantly pumped emissions from lanthanides, where a small part of the emission profiles at shorter wavelength side. Therefore the basis of the discussion in the present work is not correct (eg, the descriptions in Fig 1 caption are wrong).

Furthermore, these in-band emission depends sensitively on the temperature, and it is much obvious for observation of the anti-Stokes profile part at RT than the low temperature such as at 100 K or 4K, see Figs 3-4. These data indicate the close association of the emission with the thermalization and the need for phonons in assisting the anti-Stokes emission part. In Figs 3,4 we can also see that the main emission photons have lower energy than the pump laser, further suggesting that it is not photon upconversion.”

Our response to R1 comment 1: We believe that this detailed concern by the reviewer is due to a miscommunication of terminology. Similarly as the reviewer, we assign our anti-Stokes photoluminescence to phonon-assisted processes:

“We recently reported on anti-Stokes photoluminescence in degenerately n-doped InP nanowires,^{24,25} where the anti-Stokes mechanism relies on absorption of photons with energies less than the Fermi energy of the degenerately doped material, followed by phonon scattering of the photo-excited hole to higher k-values. Such anti-Stokes photoluminescence has so far not been reported for bulk semiconductors. In this work we demonstrate, that such phonon assisted anti-Stokes photoluminescence is not limited to nanoscale materials, but is a more general property of direct bandgap semiconductors.”

The reviewer appears to equate our previous use of “photon upconversion” with “two-photon absorption (TPA)”.

For this revision, to avoid misunderstandings, we have changed terminology from “upconversion” and “down conversion” to “anti-Stokes photoluminescence” and “Stokes photoluminescence” throughout. As in the original version, we highlight in the introduction that TPA is one process that can lead to anti-Stokes photoluminescence, but not the only process. Importantly for our later analysis, we highlight in the introduction phonon-assisted processes for anti-Stokes photoluminescence:

“Two other anti-Stokes mechanisms involve phonons. In one, the simultaneous absorption of photons and phonons enables absorption below the bandgap^{22,23} and can lead to upconverted band edge luminescence. Alternatively, photo-generated charge carriers can be excited to higher states through phonon absorption, which can lead to recombination involving these higher energetic excited states.^{24”}

Regarding the importance of our work: Anti-Stokes photoluminescence has not been reported previously in degenerately doped semiconductors. The novelty of our work is the study of such anti-Stokes photoluminescence through PL and PLE experiments at varying temperature. From the analysis of those results, we draw conclusions regarding the carrier dynamics and radiative recombination mechanism in degenerately doped direct bandgap semiconductors:

“• We find that the radiative recombination in our samples is strongly dominated by k-conserving vertical transitions, despite high doping. This finding is in contrast to the conventional view in literature where it is often assumed that the k-selection rule is violated in strongly doped semiconductors.

• We will argue that the photoluminescence lineshape is determined by the distribution of holes recombining vertically (conserving k) with the Fermi sea of electrons. In addition we find that the charge carrier recombine before the holes have thermalised to the valence band maximum. This means that the photoluminescence lineshape is not determined by holes that have relaxed to the valence band maximum before recombining, which would lead to photoluminescence peaked at the band gap.”

Commonly, it is believed in the solid state physics research community that excess carriers in degenerately doped samples thermalize to the band-edges from where they can recombine radiatively without k-conservation. Thus, our results challenge the common viewpoint. The new perspective on carrier dynamics and recombination, revealed by our study, is of importance for for a proper description of carrier dynamics and radiative recombination in degenerately doped direct bandgap semiconductors, which are one of the cornerstones in opto-electronic applications.

Reviewer 2

We are glad to hear that the reviewer finds our paper to provide an important perspective on the phenomena of up-conversion in semiconductor alloys. We are especially happy that the reviewer finds that our experiments are carefully conducted to make a strong case for our interpretation. The reviewer raises the following points to make the paper clearer and more comprehensive:

R2 comment 1: *“The authors use the terms up-conversion and down-conversion. I personally feel that these terms should correspond to either the energy of two photons being combined to produce one higher energy photon (up-conversion), or the reverse, the energy of one high energy photon being used to produce two lower energy photons. I am somewhat comfortable with the authors use of up-conversion, but not down-conversion which they seem to use to describe the conventional Stokes-shift photoluminescence. Down conversion should not conserve particle number. Sometimes this is called 'down-shifting' by the photovoltaics community, but to my mind it is simply a Stokes shift in PL. I would suggest the authors remove the cases where down-conversion is used and replace it with Stokes PL.”*

Our response to R2 comment 1: We changed the terminology to anti-Stokes PL and Stokes PL throughout the presentation of our work.

R2 comment 2: *“The authors have made a good attempt at reviewing the literature. Since their paper is quite general in its appeal, I feel they could reference a little more widely. For inorganic UC systems I am aware of the following recent papers which they may wish to consider:*

D. M. Tex, I. Kamiya, and Y. Kanemitsu, “Efficient upconverted photocurrent through an Auger process in disklike InAs quantum structures for intermediate-band solar cells,” Phys Rev B, vol. 87, no. 24, 2013.

N. P. Hylton, et al., “Photoluminescence upconversion at GaAs/InGaP2 interfaces driven by a sequential two-photon absorption mechanism,” Phys Rev B, vol. 93, no. 23, p. 235303, Jun. 2016.”

Our response to R2 comment 2: We now consider and cite these two papers as follows:

“For instance Auger assisted upconverted photocurrent has been studied for intermediate band solar cells³² and photochemical upconversion for other photovoltaic devices.^{33”}

R2 comment 3: *“Finally the authors mention the possibility for using up-conversion for solar energy conversion. The laser intensities used in the experiments here are greatly in excess of any photon density that can be achieved with sunlight so this application does not seem to be compelling. To my knowledge, molecular up-conversion systems appear to be most promising for this application, see for example the recent review: T. F. Schulze and T. W. Schmidt, “Photochemical upconversion: present status and prospects for its application to solar energy conversion,” Energy Environ. Sci., vol. 8, no. 1, pp. 103–125, 2015”*

Our response to R2 comment 3: Our main purpose with the study is to demonstrate upconversion and to present new insights into the carrier dynamics and radiative recombination in degenerately doped direct bandgap semiconductors, which are widely used for optoelectronic applications. We use the observed anti-Stokes photoluminescence (previously termed *upconversion*) to probe the carrier dynamics in a way that has not possible in previous studies. Thus, at this point, we don't propose to use the observed anti-Stokes luminescence for energy-conversion application. For this revised version, we have revised the presentation carefully to emphasize the fundamental semiconductor physics focus of our study. Specifically, we emphasize now in the conclusion that our results pertain to the materials *per se*.

Reviewer 3

The reviewer feels that our study is a bit preliminary and raises the following points for discussion:

R3 comment 1: “The authors are suggested to verify the phenomena in other materials systems and investigate this effect in a real semiconductor device.”

Our response to R3 comment 1: Our study demonstrates for the first time anti-Stokes photoluminescence in degenerately doped direct bandgap bulk semiconductors (n-type and p-type InP and GaAs). Thus, we cover two optoelectronically very important semiconductor materials systems. Furthermore, we have no reason to believe that such anti-Stokes luminescence would be exclusive to only these two materials systems among the direct bandgap semiconductors.

Importantly, the analysis of the discovered anti-Stokes luminescence revealed carrier dynamics at edge with the common viewpoint in the research field. Specifically, we conclude that:

- We find that the radiative recombination in our samples is strongly dominated by k-conserving vertical transitions, despite high doping. This finding is in contrast to the conventional view in literature where it is often assumed that the k-selection rule is violated in strongly doped semiconductors.
- We will argue that the photoluminescence lineshape is determined by the distribution of holes recombining vertically (conserving k) with the Fermi sea of electrons. In addition we find that the charge carrier recombine before the holes have thermalised to the valence band maximum. This means that the photoluminescence lineshape is not determined by holes that have relaxed to the valence band maximum before recombining, which would lead to photoluminescence peaked at the band gap.”

Commonly, it is believed that excess carriers in degenerately doped semiconductors thermalize to the band-edges from where they can recombine radiatively without k-conservation. Thus, our results challenge the common viewpoint. The new perspective on carrier dynamics and recombination, revealed by our study, is of importance for a proper description of carrier dynamics and radiative recombination in degenerately doped direct bandgap semiconductors, which are one of the cornerstones in optoelectronic applications. We highlight this point through:

“Finally, we would like to highlight that we saw in our steady-state experiment a strong indication of lack of thermalization of photogenerated minority carriers. Such an observation challenges the use of simplified models, based on the thermalization assumption, for describing carrier dynamics in the analysis of semiconductor devices with highly doped regions, such as degenerately doped p⁺n⁺-junction based solar cells.”

We would like to point out that our study is a fundamental semiconductor physics study, which aims to unveil new details about the carrier dynamics in commonly used semiconductor systems. We used the well-controlled system consisting of a homogeneously doped bulk sample. In contrast, a study of a specific semiconductor device would introduce additional uncertainties of the sample configuration, such as doping profile and spatially varying materials quality due to varying conditions during the fabrication, which would cast uncertainty to the origin of observed behavior. Thus, a study of a full semiconductor device would require a substantial effort with a large set of control samples to draw definite conclusions.

Therefore, our fundamental results on the carrier dynamics in a well-controlled materials system are the starting point to ignite further studies towards device applications based on such materials systems. Importantly, a more accurate description of carrier dynamics would have implications for the design and optimization of devices based on degenerately doped semiconductors.

R3 comment 2: “The study also looks too specialized to the referee. To date, a number of materials systems including nanocrystals and small molecules are available for upconverting photon energy, inspiring quite a few new applications. This study focuses on a specific physical process in a particular material, and the implications of the investigation are not clear. It may not appeal to the broad readership of nature communications or excite the direct interests of the community.”

Our response to R3 comment 2: We would like to kindly argue against the reviewers concern about the importance of our study. We present a fundamental study of the carrier dynamics in degenerately doped direct bandgap bulk semiconductors, which are used widely in both academic research and in the optoelectronic industry.

Thus, our study brings additional fundamental insight into a widely used materials system. From the application point of view, any added understanding of the carrier dynamics could have broad implications. For example, a better understanding of the semiconductor physics, like carrier dynamics, could enable enhanced design and optimization of devices.

Reviewers' Comments:

Reviewer #1:

Remarks to the Author:

At first time upconversion based on semiconductor materials have been observed and studied. After careful reading of the manuscript I came to the conclusion that the results as well as the way they are presented meets the requirements of a high impact journal

The changes the authors made to the manuscript are convincing to me. I had a look on the comments of the other referees and feel that (against my first impression) the paper is suitable for publication in Nature Communications

Reviewer #3:

Remarks to the Author:

The referee is not satisfied with the response of the authors. The authors emphasize that the anti-Stokes luminescence is a general attribute of direct semiconductors and their findings would have important implications in the optoelectronic industry. The referee is keen to see a demonstration in an optoelectronic device.